# Prevalence and Associated Factors of Compliance Behaviors among Middle-Aged and Older Hypertensive Patients in China: Results from the China Health and Retirement Longitudinal Study

**DOI:** 10.3390/ijerph17197341

**Published:** 2020-10-08

**Authors:** Jianjian Liu, Ying Yang, Jiayi Zhou, Tianyu Liu, Wenjie Zhang, Liuyi Wei, Shaotang Wu

**Affiliations:** 1School of Health Sciences, Wuhan University, Wuhan 430071, China; jianjianliu@whu.edu.cn (J.L.); yangying@whu.edu.cn (Y.Y.); 2017302180144@whu.edu.cn (J.Z.); tianyuliulty@whu.edu.cn (T.L.); wenjiezhang@whu.edu.cn (W.Z.); 2018283050051@whu.edu.cn (L.W.); 2Global Health Institute, Wuhan University, Wuhan 430072, China

**Keywords:** patient compliance, medication adherence, blood pressure monitoring, hypertension, middle-aged and older adults, Health Ecology Model

## Abstract

Partial or total non-adherence has been recognized as major issues in the long-term management of hypertension. This study aims to investigate the prevalence and associated factors of compliance behaviors among Chinese middle-aged and older hypertensive patients. A sample of 6308 hypertensive patients aged ≥45 years was obtained from the 2015 China Health and Retirement Longitudinal Study (CHARLS) data. Two compliance behaviors were involved including medication and blood pressure monitoring. Stratified binary logistic regression analysis was employed to examine the associated factors. 77.2% of the participants reported medication compliance, and 40.7% complied with blood pressure monitoring. Better medication compliance associated with older age, overweight or obesity, one or ≥3 complications, no drinking, living in urban areas, and health education. Better blood pressure monitoring compliance associated with older age, overweight or obesity, ≥3 complications, normal activities of daily living (ADL), no smoking, sleep duration of 6–8 h, better cognitive function, living in urban areas, education level of middle school or above, and health education. Chinese middle-aged and older hypertensive patients experienced unoptimistic compliance behaviors, especially for blood pressure monitoring. Special attention and targeted interventions are urgent for the high-risk population of poor compliance behaviors, such as rural individuals, low educational population, and younger hypertensive patients.

## 1. Introduction

Population ageing has become an irreversible phenomenon worldwide. China is one of the most ageing countries with the number of people aged 65 and above ranking first in the world [1]. By the end of 2019, there were 175.99 million older adults aged 65 years and older in China, accounting for 12.6% of the total Chinese population [2]. With the aggravation of population ageing, chronic non-communicable diseases represented mainly by cardiovascular and cerebrovascular diseases have become a major threat to human health, especially for people in older age [3]. According to the Chinese Center for Disease Control and Prevention (China CDC), the prevalence of chronic diseases among Chinese older adults aged 60 years and older reached 76.3% in 2019, among which hypertension had become the most prevalent disease, its prevalence being 58.3% [4]. However, while more prevalent among older adults, an increasing incidence of hypertension in the younger population is observed [5]. Li’s [6] study investigation between 1991 and 2001 found that the prevalence of hypertension increased by 12.92%, 14.94%, and 6.36% in participants aged 18–44 years, 45–64 years, and ≥65 years, respectively, indicating a younger trend of the incidence of hypertension. With the feature of long duration, high mortality and disability rate, hypertension brings huge challenges to not only the physical and mental health of middle-aged and older people, but health service system, and socio-economic development [7].

Patient compliance generally refers to the degree to which the patient’s behavior (medication, diet, lifestyle modification, exercise, etc.) conforms to the clinical doctor’s prescription or advice [8]. Previous studies have proposed that poor compliance behaviors are associated with disease progression, treatment failure, and multiple hospital admissions, which seriously threatens the quality of life of people with chronic illness [9]. A three-year follow-up study found that hypertensive patients with poor compliance behaviors were more likely to suffer from other chronic diseases, such as coronary disease, cerebrovascular disease, and chronic heart failure [10]. Kim et al.’s finding [11] from a sample of 564,782 hypertensive patients indicated that as compared with individuals who had a medication possession ratio (MPR) of ≥80%, the risks of comorbidity were 2.01 and 1.36 times higher in those with <20% MPR and 40–59% MPR, respectively. The results supporting that medication compliance behaviors can effectively reduce the risk of comorbidity. Compliance behaviors played important roles in effectively controlling blood pressure, reducing complications, and improving quality of life in patients with hypertension [12]. Medication is the most direct and effective way to control blood pressure; thus, medication compliance was always the most sensitive and commonly used indicator for compliance behaviors of patients with hypertension [13]. Blood pressure monitoring is fundamental for the evaluation of disease condition and the observation of therapeutic effect, which was proposed to be implemented regularly by international education guidelines for hypertensive patients [14,15]. Previous studies reported the significant effectiveness of blood pressure monitoring on blood pressure control of hypertensive patients. Huang et al. [16] found that blood pressure monitoring can improve the effect of blood pressure control by significantly reducing the systolic and diastolic blood pressure of hypertensive patients. Besides, complying with lifestyle modifications was also the important content in hypertension management. However, it is difficult to determine if individuals have received health guidance for lifestyle modification from their doctors, as well as to assess the degree of their compliance with doctors’ advice on smoking, drinking, and exercising. Therefore, this study considered the two important and measurable behaviors-medication and blood pressure monitoring, as indicators for compliance behaviors of hypertensive patients.

Previous studies conducted in different regions [17], different age groups [18], and different economic levels [19] consistently found that hypertensive patients experienced poor compliance behaviors in China. As for medication compliance behavior, a survey by Zhang et al.’s [20] found that Chinese hypertensive patients have poor medication compliance, about 24.7%-55.1%. World Health Organization (WHO) [21] reported that only 43% of hypertensive patients can insist on taking antihypertensive drugs in China. For blood pressure monitoring behavior, Xu et al.’s survey [22] in rural areas of Beijing found that only 12.0% of the hypertensive patients insisted on monitoring blood pressure regularly. Peng et al. [23] reported a proportion of 38.5% for persisting in measuring blood pressure in the morning and evening among hypertensive patients in Nantong. Although previous studies to a certain extent reflected a frustrating situation of compliance behaviors among Chinese hypertensive patients, they were mostly limited to a certain region, a certain age group, or a certain behavior, and the research at the national level is extremely scarce. What is more, with the increasing prevalence of chronic diseases in the middle-aged population [24], the compliance behaviors of middle-aged hypertensive patients should not be ignored.

Individuals’ compliance behaviors are usually influenced by multiple factors. Previous studies have reported the correlations between compliance behaviors and gender [25], health knowledge [26], and socio-economic factors [27] among individuals with hypertension. However, these studies were generally limited to a certain region and most focused only on medication compliance. Knowledge is limited regarding the related factors of compliance behaviors of hypertensive patients in China, especially for the behavior of blood pressure monitoring.

In July 2019, the Chinese government implemented “Healthy China Action (2019–2030)” [28], in which self-management of chronic diseases was emphasized as the core strategy. Therefore, to clarify the above issues, and to provide references for the development of targeted interventions towards health management on hypertension in China, we conducted this study. Using nationally representative data, this study aims to (a) describe the prevalence of compliance behaviors among Chinese middle-aged and older hypertensive patients and (b) examine the associated factors of compliance behaviors.

## 2. Materials and Methods

### 2.1. Data Sources

This study is designed as a cross-sectional study. We used data from the China Health and Retirement Longitudinal Study (CHARLS), a longitudinal cohort survey conducted by National School of Development, Peking University. In this survey, Chinese adults aged 45 years and older were investigated adopting a multi-stage stratified cluster sampling method [29]. CHARLS aimed at collecting a high quality nationally representative data to serve the needs of analysis on population ageing and interdisciplinary research on older adults. The survey has been conducted since 2011 and followed up every two or three years. The latest available national survey data of CHARLS was fielded in 2015, covering more than 10,000 households and 17,500 individuals in 150 counties/districts of 28 provinces [30].

### 2.2. Study Population

This study used the 2015 CHARLS data (*n* = 21,097). Middle-aged and older individuals diagnosed with hypertension were included as study subjects (*n* = 6424). Participants aged ≥45 years were selected based on the WHO age criteria for middle-aged and older people. Hypertension was identified based on participants’ self-report, i.e., respondents answered “yes” in the following two questions were recognized as hypertensive patients: “Have you been diagnosed with hypertension by a doctor?” and “Do you know if you have hypertension?” We excluded subjects with missing information on compliance behaviors and predictor variables listed below (*n* = 116). Finally, 6308 individuals with hypertension were included in the analyses.

### 2.3. Description of Measures

#### 2.3.1. Compliance Behaviors

In this study, compliance behaviors consisted of taking medicine and monitoring blood pressure. Medication behavior was measured by the question “Are you now taking any of the following treatments to treat or control your hypertension: taking Chinese traditional medicine, taking Western modern medicine, or none of the above?” The answers “taking Chinese traditional medicine” or “taking Western modern medicine” were considered as positive medication behavior, i.e., compliance with medication. Blood pressure monitoring behavior was assessed by asking “During last year (last 12 months), how many times have you had blood pressure examination?” According to the health management guideline for hypertension [31], hypertensive patients are recommended to monitor their blood pressure at least once a month. Therefore, we dichotomized the number of blood pressure monitoring into <12 times/year and ≥12 times/year. Monitoring of blood pressure ≥12 times/year was considered as positive behavior, i.e., compliance with blood pressure monitoring.

#### 2.3.2. Predictor Variables

This study applied Health Ecology Model to interpret the associated factors of compliance behaviors, which was widely applied in health management of chronic diseases [32]. The model holds that individual or group health is not only affected by individual factors, but also by social interpersonal networks; living and working environment; and macro-social and policy environments such as economy, culture, and health systems [33]. Based on Health Ecology Model, social determinants of health were categorized by Helnk et al. [34] as downstream factors, midstream factors, and upstream factors. Downstream factors directly influence the individuals’ health and include genetic factors and biological and biochemical factors, such as family history and personal physiological indicators. Midstream factors include health-related behaviors and psychological factors, such as eating, exercising, smoking, and drinking, which are the intermediate factors that affect health. Upstream factors are overarching social determinants influencing health, which cover environment, society, economics, culture, politics, and medical insurance, and include working environment, living conditions, social insurance, national or local policies, and political and economic system. According to this classification method, we divided factors potentially related to compliance behaviors into three levels: downstream, midstream, and upstream factors.

Downstream factors included age, sex, body mass index (BMI), number of complications (hypertension was excluded), and activities of daily living (ADL). Individuals’ ADL ability were measured through six basic activities of daily living (BADL) and six instrumental activities of daily living (IADL), including toileting, eating, dressing, controlling urination and defecation, getting out of bed, bathing, doing household chores, preparing hot meals, shopping, making phone calls, taking medications, and managing assets. In this study, participants who reported impaired function in any one of the 12 activates were considered as ADL decline.

Midstream factors consisted of smoking, drinking, regular exercise, sleep duration, depression, and cognitive function. Depression was assessed by the modified 10-Item Center for Epidemiologic Studies Depression Scale (CESD-10) [35]. The scale comprises 8 negative questions and 2 positive ones and each question includes 4 options: rarely or none of the time (<1 day), some or a little of the time (1–2 days), occasionally or a moderate amount of the time (3–4 days), and most or all of the time (5–7 days). Positive items scored 0 to 3, and negative items scored 3 to 0 from rarely to most. The total score of CESD-10 ranges from 0 to 30, and participants who scored ≥10 were considered as having a depressed mood [36]. The CESD-10 has demonstrated high validity and reliability among Chinese population [37]. Cognitive function was measured through orientation, attention, memory, and visual spatial ability, in which the total score ranged from 0 to 21 [38]. In the present study, we categorized participants’ cognitive function as “scored <10” and “scored ≥10”.

Upstream factors included marital status, place of residence, educational level, personal income, medical insurance, and health education. The detailed information of all the above variables, including the source of items in the CHARLS 2015 Questionnaire, re-coding, classification, or definition, are summarized in Appendix A.

### 2.4. Statistical Analysis

Data analyses were performed by using the Statistical Package for the Social Sciences (SPSS) version 21.0 (SPSS Inc., Chicago, IL, USA), with a significance level of 0.05. Descriptive statistics were employed to describe the characteristics of the participants and the condition of compliance behaviors. Chi-square test was used to compare the differences of medication and blood pressure monitoring among participants with different downstream, midstream, and upstream factors. Variables with *p*-value <0.05 in chi-square tests were selected as independent variables for regression models. Compliance behaviors, i.e., compliance with medication and compliance with blood pressure monitoring, were set as dependent variables, respectively. For each dependent variable, 4 models were established using stratified unconditional binary logistic regression analysis. Firstly, downstream factors with significant difference in chi-square tests were included as reference model (model 1). Secondly, the midstream and upstream factors with significance in chi-square tests were included in model 2 and model 3, respectively, in which downstream factors were controlled to compare the effect of different factors on compliance behaviors. Finally, all the independent variables with significant differences in chi-square tests were included in the same model (model 4). Hosmer–Lemeshow test was applied to evaluate the goodness of fit of each model.

### 2.5. Ethical Statements

This study used de-identified and publicly available data from the CHARLS website (http://charls.pku.edu.cn/zh-CN/page/data/2015-charls-wave4). Protocols, instruments, and the process used to obtain informed consent in CHARLS were approved by the Biomedical Ethics Review Committee of Peking University (IRB00001052-11015) [39]. All participants provided written informed consent for their participation in the survey.

## 3. Results

### 3.1. General Information

A total of 6308 middle-aged and older individuals with hypertension were involved in this study, with an average age of 63.3 years (SD = 9.8). Among the participants, 53.1% (*n* = 3350) were female, 41.8% (*n* = 2634) aged 65 years and older, 82.8% (*n* = 5220) were married, 75.8% (*n* = 4781) received less than 6 years of education, 70.5% (*n* = 4447) lived in rural areas, and 77.9% (*n* = 4917) had one or more complications. Details on the demographics characteristics of the participants are listed in Table 1.

With regards to compliance behaviors, 77.2% (*n* = 4870) of the participants reported compliance with medication, and 40.7% (*n* = 2567) reported compliance with blood pressure monitoring. Besides, the median frequency of blood pressure monitoring was 7 times/year in the present sample (*n* = 145). The most frequency of blood pressure monitoring was 3 times/year (*n* = 625), accounting for 9.9% of the total participants.

### 3.2. Univariate Analysis

The results of chi-square tests are presented in Table 1. Significant differences were observed in medication compliance among participants with different sex, age, BMI, number of complications, ADL, smoking, drinking, depression, cognitive function, place of residence, educational level, personal income, and health education (all *p*-values < 0.05). The behavior of compliance with blood pressure monitoring significantly differed in participants with different sex, BMI, number of complications, ADL, smoking, drinking, depression, cognitive function, place of residence, educational level, personal income, and health education (all *p*-values < 0.05).

### 3.3. Multi-Factor Regression Analysis

#### 3.3.1. Compliance with Medication Behavior

Table 2. demonstrates the results of logistic regression analysis for medication compliance behavior. In the final model 4, when compared with 45–54 age group, participants aged 55–64 (*OR* = 1.57, 95% CI = 1.34–1.84), 65–74 (*OR* = 1.89, 95% CI = 1.58–2.25), and ≥75 (*OR* = 2.25, 95% CI = 1.77–2.85) reported better compliance with medication. Compared with normal BMI, overweight (*OR* = 1.44, 95% CI = 1.25–1.65) and obese (*OR* = 1.72, 95% CI = 1.41–2.09) participants reported higher proportion of medication compliance. Compared with participants with no complication, those with one (*OR* = 1.20, 95% CI = 1.02–1.42) and ≥3 (*OR* = 1.65, 95% CI = 1.37–1.98) had better compliance with medication. Individuals who received health education (*OR* = 1.98, 95% CI = 1.75–2.24) were more likely to report medication compliance. Participants who drank (*OR* = 0.71, 95% CI = 0.61–0.82) or lived in rural areas (*OR* = 0.85, 95% CI = 0.74–0.99) reported a lower proportion of compliance with medication. Hosmer–Lemeshow test indicated good fitness of the model (*χ*^2^ = 13.74, *p*-value > 0.05).

#### 3.3.2. Compliance with Blood Pressure Monitoring Behavior

Table 3 presents the results of logistic regression analysis for blood pressure monitoring compliance behavior. In model 4, when compared with 45–54 age group, participants aged 55–64 (*OR* = 1.16, 95% CI = 1.00–1.34), 65–74 (*OR* = 1.29, 95% CI = 1.11–1.51), and ≥75 (*OR* = 1.49, 95% CI = 1.22–1.81) were more likely to comply with blood pressure monitoring. Compared with normal weight, overweight (*OR* = 1.25, 95% CI = 1.11–1.42) and obese (*OR* = 1.27, 95% CI = 1.07–1.49) participants showed higher proportion of compliance with blood pressure monitoring. Participants with ≥3 complications reported better blood pressure monitoring compliance behavior than those with no complication (*OR* = 1.39, 95% CI = 1.18–1.62). Participants with cognitive function score ≥10 (*OR* = 1.29, 95% CI = 1.14–1.46) and who received health education (*OR* = 1.29, 95% CI = 1.16–1.44) displayed a higher proportion of compliance with blood pressure monitoring. Compared with no formal education or illiterate, participants who received education level of middle school (*OR* = 1.98, 95% CI = 1.58–2.49) and high school or above (*OR* = 1.29, 95% CI = 1.16–1.44) had a better compliance behavior. However, the odds ratios of people who declined in ADL, smoked, slept less than 6 h and lived in rural areas were 0.845 (95% CI = 0.75–0.95), 0.758 (95% CI = 0.66–0.87), 0.874 (95% CI = 0.78–0.99), and 0.610 (95% CI = 0.54–0.69), respectively. Hosmer–Lemeshow test showed the models fitted well (*χ*^2^ = 6.46, *p*-value > 0.05).

## 4. Discussion

Compliance behavior is a core connotation and an important premise of the self-management of hypertension. In this study, 77.2% of the participants reported compliance with medication, which is much higher than previous findings in China and slightly lower than Al-Daken and Eshah’s [40] study in developed countries (82.8%). The differences might be due to the distinction of study population and assessment method of medication compliance. It is worth noting that this study investigated individuals’ medication compliance behavior by simply asking if the participants were taking any medicine to treat or control hypertension, rather than assessing whether the medication was strictly in accordance with the doctor’s prescription or advice. Thus, there is a possibility that this study may overestimate the prevalence of medication compliance behavior.

This study found that 40.7% of the participants reported compliance with blood pressure monitoring, i.e., measuring blood pressure ≥12 times/year. The highest frequency of blood pressure monitoring was 3 times/year (9.9%), with the median frequency of 7 times/year for blood pressure monitoring. The results indicated that the frequency of blood pressure monitoring in Chinese middle-aged and older patients did not meet the recommendations of hypertension prevention and treatment guidelines. Insufficient blood pressure monitoring has become an important loophole in hypertension health management [41]. Moreover, the present finding was far lower than that in Japan [42] and the United States [43]. It is suggested that health education and guidance on blood pressure monitoring should be strengthened among Chinese middle-aged and older population, and the implementation of family blood pressure monitoring should be promoted.

The results of multivariate analysis indicated that, among the downstream factors, age, BMI, number of complications, and ADL were associated with individuals’ compliance behaviors. Compared with the 45–54 age group, individuals aged 55 years and older reported better compliance behavior in this study, which is consistent with Su et al. [44] and Xiao et al. [45]. This may be due to the fact that older people generally paid more time and energy to their health and had stronger willingness to comply with doctors. We found better compliance behaviors in overweight and obese individuals than those with normal weight, which is in line with Reaven’s [46] finding. This might be explained as that overweight or obese people are likely to suffer from other diseases and thus pay more attention to their health. This study found that participants with ≥3 complications had significantly better compliance behaviors. Abinaya and Aanandhi’s [47] research in diabetes mellitus presented a similar finding. Individuals complicated with multiple diseases experience poor health status and are more eager to improve their health by complying with their doctors and health educators. Further analysis found that the most common complications in this sample were chronic diseases, such as arthritis or rheumatism, heart disease, and dyslipidemia. Follow-up studies are needed to explore the relationship between the types of complications and compliance behavior among hypertensive patients. Besides, participants with declined ADL function demonstrated poor compliance behavior in blood pressure monitoring in this study, which might be mainly due to the limited ability of daily living. In summary, at the downstream level of health ecological model, the compliance behaviors of hypertensive patients with younger age, normal weight, without complications, and declined ADL function should be emphasized.

This study found that, in the midstream level, smoking, drinking, sleep duration, and cognitive function were significantly associated with compliance behaviors. Compliance behavior has been identified to be correlated with individuals’ health-related behaviors [48]. As common health-related behaviors, smoking, drinking, and sleep duration were measured in this study, and the results confirmed the negative effect of unhealthy lifestyles on patients’ compliance, which is consistent with Heymann et al. [49] The results of regression analysis also indicated that hypertensive patients with cognitive scores ≥10 were more likely to comply with blood pressure monitoring. Therefore, when it comes to compliance behavior interventions, hypertensive patients who smoke, drink, sleep poorly, or with cognitive scores <10 should be concerned, and the guidance of lifestyle modification should be promoted.

We also found that, in the upstream level, place of residence, education level, and health education were related to compliance behaviors of hypertensive patients. In this study, participants who lived in rural areas were less likely to comply with medication than those who lived in urban areas, which is consistent with a population-based study in Taiwan [50]. Generally, rural population had relatively low levels of income, education, and health service accessibility and utilization [51]. The great socio-economic differences between urban and rural areas will cause significant differences in individuals’ health-related behaviors. Individuals with higher socio-economic status have better awareness of hypertension prevention and control and can better insist on antihypertension therapy [38]. Besides, on the one hand, it is difficult to carry out health education on hypertension in rural areas due to resident’s limited knowledge level and health literacy. On the other hand, few hypertensive patients possess sphygmomanometer in rural areas due to the low level of economy, which also limits home self-monitoring of blood pressure [52]. In this study, patients who received health education were 1.97 times more likely to comply with medication, and 1.29 times more likely to comply with blood pressure monitoring, indicating that health education might significantly improve the compliance behavior of hypertensive patients. This fits well with Li et al.’s [53] meta-analysis in China and Al-Rubaey and Shwaish’s survey [54] in Baghdad. We detected significantly better blood pressure monitoring behavior in participants with ≥12 years of education than in those with no formal education, which is in good agreement with Wang et al.’s [55] community-level survey in Chengdu, China. From the upstream level, the accessibility of health management services might have a positive effect on compliance behaviors, suggesting that the health management of chronic diseases should be improved by the construction of family-doctor system. The primary health service institutions should implement the health management services for patients with hypertension and focus on improving the health literacy of rural patients.

To promote health management of chronic diseases, Chinese government implemented “Healthy China Action (2019–2030)” in July 2019, in which “Prevention and Treatment of Cardiovascular and Cerebrovascular Diseases” was involved as one of the fifteen major actions. Improving compliance behaviors is an important and effective approach to promote the self-management of hypertension patients. On the individual level, targeted health education should be implemented in middle-aged and older hypertensive patients, such as weight control, exercise and diet guidance, and quitting smoking, so as to improve knowledge, enhance awareness, and then promote the change of behaviors and lifestyle. Special attention should be paid to the population who were more likely to have poor compliance behaviors, such as hypertensive patients who were middle-aged, had complications, or drank. At the community level, training on blood pressure monitoring should be provided regularly for hypertensive patients, especially for low socioeconomic status population. Family blood pressure monitoring should be promoted to encourage regular monitoring of blood pressure. It is recommended to further promote the implementation of a Family-Doctor System in China, so as to improve the accessibility of blood pressure monitoring. Besides, it would be helpful if government provided financial assistance for the purchase of blood pressure monitors.

Several limitations should be mentioned regarding the present study. Firstly, all indicators applied in this study were obtained through participants’ self-report; thus, they are likely to bring biases. Secondly, there might be a certain degree of inadequacy regarding the definition of compliance behaviors in this study. For example, we only included medication and blood pressure monitoring as two indicators to evaluate patients’ compliance behaviors. Medication compliance behavior was assessed by simply asking if the participants were taking any medicine to treat or control hypertension, which may lead to overestimation of the results. And we did not take into account the amount of medicine or the severity of the diseases. The number of blood pressure monitoring only depended on recall, and the monitor place (hospital or home) was not clear. Thirdly, the cross-sectional nature of this study may be considered as a limitation, since no causal inferences can be drawn from the results. Despite these limitations, this study used nationally representative data to systematically examine the condition and associated factors of compliance behaviors among Chinese hypertensive patients. The results might be a valuable reference for the health management practices of hypertension. What is more, this study contributed to use the Health Ecological Model to explore the effect of multilevel factors on compliance behaviors, providing references for targeted prevention. However, considering the limitations in the present study, future studies should put emphasis on study design and comprehensively focus on hypertensive patients’ compliance behaviors (including lifestyle).

## 5. Conclusions

In this study, 77.2% of the middle-aged and older hypertensive patients reported medication compliance, and 40.7% reported compliance with blood pressure monitoring, i.e., measuring blood pressure ≥12 times/year. Medication compliance behavior was better among patients who were aged, were overweight or obese, had one or ≥3 complications, received health education, did not drink, and lived in urban areas. Better blood pressure monitoring compliance behavior was associated with older age, overweight or obesity, having ≥3 complications, normal ADL ability, no smoking, appropriate sleep duration, better cognitive function, living in urban areas, higher education lever, and receiving health education. It is necessary to develop targeted interventions towards complication behaviors of hypertensive patients in China, especially for blood pressure monitoring. Special attention should be paid to the high-risk population of poor compliance behaviors, such as rural individuals, low educational population, and younger hypertensive patients. Moreover, supporting policies regarding hypertension management, including popularizing health knowledge, reducing health risk behaviors, and strengthening family blood pressure monitoring, etc., might make great sense.

## Figures and Tables

**Table 1 ijerph-17-07341-t001:** Compliance behaviors in participants with different demographic characteristics.

Variable	Categories	All Participants	Medication Compliance		Blood Pressure Monitoring Compliance	
		*n* (%)	No (%)	Yes (%)	No (%)	Yes (%)
Sex	Male	2958 (46.9)	26.2	73.8 ***	57.7	42.3 *
	Female	3350 (53.1)	20.0	80.0	60.8	39.2
Age (years)	45–54	1381 (21.9)	31.2	68.8 ***	60.2	39.8
	55–64	2293 (36.4)	22.6	77.4	59.8	40.2
	65–74	1783 (28.3)	19.3	80.7	58.6	41.4
	≥75	851 (13.5)	17.6	82.4	58.2	41.8
BMI (kg/m^2^)	Normal (18.5–23.9)	1791 (28.4)	27.9	72.1 ***	65.0	35.0 ***
	Underweight (<18.5)	187 (3.0)	23.5	76.5	69.0	31.0
	Overweight (24–27.9)	3266 (51.8)	21.6	78.4	56.5	43.5
	Obese (≥28)	1064 (16.9)	18.1	81.9	56.6	43.4
Number of complications	0	1391 (22.1)	29.3	70.7 ***	63.0	37.0 ***
	1	1764 (28.0)	24.5	75.5	60.4	39.6
	2	1381 (21.9)	23.4	76.6	60.5	39.5
	≥3	1772 (28.1)	15.8	84.2	54.4	45.6
ADL	Normal	3844 (60.9)	25.2	74.8 ***	56.8	43.2 ***
	Declined	2464 (39.1)	19.2	80.8	63.3	36.7
Smoking	No	4581 (72.6)	20.9	79.1 ***	58.2	41.8 **
	Yes	1727 (27.4)	28.1	71.9	62.4	37.6
Current drinking	No	4298 (68.1)	19.7	80.3 ***	58.8	41.2
	Yes	2010 (31.9)	29.6	70.4	60.5	39.5
Regular exercise	No	4032 (63.9)	22.6	77.4	59.1	40.9
	Yes	2276 (36.1)	23.4	76.6	59.7	40.3
Sleep duration	<6 h	1944 (30.8)	21.1	78.9	62.2	37.8 ***
	6–8 h	3806 (60.3)	23.6	76.4	57.1	42.9
	≥9 h	558 (8.8)	23.8	76.2	64.7	35.3
Depressive mood	No	4153 (65.8)	24.1	75.9 **	57.4	42.6 ***
	Yes	2155 (34.2)	20.5	79.5	63.1	36.9
Cognitive function score	<10	2573 (40.8)	21.3	78.7 *	66.8	33.2 ***
	≥10	3735 (59.2)	23.9	76.1	54.2	45.8
Marital status	Non-married	1088 (17.2)	22.4	77.6	61.9	38.1
	Married	5220 (82.8)	23.0	77.0	58.8	41.2
Educational level	0	1528 (24.2)	19.8	80.2 **	67.4	32.6 ***
	6 years	3253 (51.6)	23.8	76.2	60.3	39.7
	9 years	950 (15.1)	22.9	77.1	54.4	45.6
	≥10 years	577 (9.1)	25.8	74.2	40.2	59.8
Place of residence	Urban	1861 (29.5)	20.5	79.5 **	46.5	53.5 ***
	Rural	4447 (70.5)	23.9	76.1	64.7	35.3
Personal income	No	4209 (66.7)	21.9	78.1 **	61.7	38.3 ***
	Yes	2099 (33.3)	24.9	75.1	54.5	45.5
Medical insurance	No	447 (7.1)	26.0	74.0	63.5	36.5
	Yes	5861 (92.9)	22.6	77.4	59.0	41.0
Health education	No	2676 (42.4)	29.4	70.6 ***	64.2	35.8 ***
	Yes	3632 (57.6)	18.1	81.9	55.7	44.3

Note: * *p*-value < 0.05, ** *p*-value < 0.01, *** *p*-value < 0.001. BMI—body mass index; ADL—activities of daily living.

**Table 2 ijerph-17-07341-t002:** Multi-factor logistic regression models for medication compliance.

Variable	Model 1 *OR* (95% CI)	Model 2 *OR* (95% CI)	Model 3 *OR* (95% CI)	Model 4 *OR* (95% CI)
**Downstream Factors**				
Sex (vs. Male)				
Female	1.33 (1.18–1.50) ***	1.08 (0.93–1.26)	1.41 (1.24–1.61) ***	1.14 (0.97–1.33)
Age (vs. 45–54)				
55–64	1.55 (1.33–1.81) ***	1.53 (1.31–1.78) ***	1.60 (1.37–1.87) ***	1.57 (1.34–1.84) ***
65–74	1.92 (1.62–2.28) ***	1.86 (1.57–2.22) ***	1.96 (1.65–2.34) ***	1.89 (1.58–2.25) ***
≥75	2.23 (1.78–2.79) ***	2.11 (1.68–2.66) ***	2.39 (1.89–3.02) ***	2.25 (1.77–2.85) ***
BMI (vs. Normal)				
Underweight	1.06 (0.74–1.52) ***	1.07 (0.75–1.54) ***	1.05 (0.73–1.51) ***	1.07 (0.74–1.54)
Overweight	1.50 (1.31–1.72) ***	1.49 (1.30–1.71) ***	1.45 (1.26–1.66) ***	1.44 (1.25–1.65) ***
Obese	1.89 (1.56–2.29) ***	1.85 (1.53–2.25) ***	1.75 (1.44–2.13) ***	1.72 (1.41–2.09) ***
Number of complications (vs. 0)				
1	1.24 (1.06–1.46) **	1.23 (1.05–1.45) *	1.21 (1.03–1.43) *	1.20 (1.02–1.42) *
2	1.25 (1.05–1.49) *	1.24 (1.04–1.47) *	1.18 (1.00–1.41)	1.16 (0.97–1.39)
≥3	1.89 (1.58–2.27) ***	1.83 (1.53–2.20) ***	1.71 (1.42–2.05) ***	1.65 (1.37–1.98) ***
ADL (vs. Normal)				
Declined	1.09 (0.95–1.24)	1.04 (0.90–1.20)	1.09 (0.95–1.25)	1.04 (0.90–1.20)
**Midstream Factors**				
Smoking (vs. No)				
Yes		0.89 (0.77–1.04)		0.87 (0.75–1.02)
Current drinking (vs. No)				
Yes		0.72 (0.62–0.83) ***		0.71 (0.61–0.82) ***
Depressive mood (vs. No)				
Yes		1.08 (0.94–1.24)		1.09 (0.95–1.26)
Cognitive function score (vs. <10)				
≥10		1.01 (0.88–1.15)		0.95 (0.82–1.09)
**Upstream Factors**				
Residence location (vs. Urban areas)				
Rural areas			0.86 (0.75–0.99) *	0.85 (0.74–0.99) *
Educational level (vs. 0)				
6 years			0.94 (0.80–1.11)	0.96 (0.81–1.14)
9 years			1.06 (0.85–1.33)	1.09 (0.87–1.37)
≥12 years			0.84 (0.65–1.08)	0.88 (0.67–1.14)
Personal income (vs. No)				
Yes			0.90 (0.79–1.02)	0.93 (0.81–1.06)
Health education (vs. No)				
Yes			1.95 (1.72–2.20) ***	1.97 (1.75–2.24) ***
Hosmer–Lemeshow test	*χ*^2^ = 4.02 *p* = 0.855	*χ*^2^ = 5.21 *p* = 0.734	*χ*^2^ = 13.37 *p* = 0.100	*χ*^2^ = 13.74 *p* = 0.089

Note: * *p*-value < 0.05, ** *p*-value < 0.01, *** *p*-value < 0.001. *OR*—odds ratio; CI—confidence interval; BMI—body mass index; ADL—activities of daily living. Model 1 was the crude model. Model 2 adjusted for the potential predictors (Regular exercise, Sleep duration), Model 3 adjusted for the potential predictors (Marital status, Medical insurance), Model 4 adjusted for the potential predictors (Regular exercise, Sleep duration, Marital status and Medical insurance).

**Table 3 ijerph-17-07341-t003:** Multi-factor logistic regression models for blood pressure monitoring compliance behavior.

Variable	Model 1 *OR* (95% CI)	Model 2 *OR* (95% CI)	Model 3 *OR* (95% CI)	Model 4 *OR* (95% CI)
**Downstream Factors**				
Sex (vs. Male)				
Female	0.89 (0.80–0.99) *	0.86 (0.76–0.97) *	1.03 (0.92–1.15)	0.94 (0.83–1.07)
Age (vs. 45–54)				
55–64	1.08 (0.94–1.24)	1.13 (0.98–1.30)	1.14 (0.99–1.32)	1.16 (1.00–1.34) *
65–74	1.18 (1.01–1.36)*	1.25 (1.08–1.46) **	1.28 (1.09–1.49) **	1.29 (1.11–1.51) **
≥75	1.35 (1.13–1.63) **	1.45 (1.20–1.75) ***	1.48 (1.22–1.80) ***	1.49 (1.22–1.81) ***
BMI (vs. Normal)				
Underweight	0.81 (0.58–1.12)	0.84 (0.61–1.18)	0.87 (0.62–1.22)	0.89 (0.64–1.24)
Overweight	1.45 (1.28–1.63) ***	1.36 (1.20–1.53) ***	1.30 (1.15–1.47) ***	1.25 (1.11–1.42) ***
Obese	1.46 (1.24–1.71) ***	1.37 (1.16–1.60) ***	1.32 (1.12–1.56) **	1.27 (1.07–1.49) **
The number of complications (vs. 0)				
1	1.15 (1.00–1.34)	1.15 (1.00–1.34)	1.11 (0.95–1.28)	1.11 (0.96–1.29)
2	1.16 (1.00–1.36)	1.15 (0.98–1.34)	1.08 (0.92–1.27)	1.08 (0.92–1.27)
≥3	1.58 (1.36–1.83) ***	1.55 (1.33–1.81) ***	1.38 (1.18–1.61) ***	1.39 (1.18–1.62) ***
ADL (vs. Normal)				
Declined	0.69 (0.61–0.77) ***	0.76 (0.68–0.86) ***	0.81 (0.72–0.91) **	0.85 (0.75–0.95)**
**Midstream Factors**				
Smoking (vs. No)				
Yes		0.75 (0.66–0.86) ***		0.76 (0.66–0.87) ***
Sleep duration (vs. 6–8 h)				
<6 h		0.88 (0.78–0.99) *		0.87 (0.78–0.99) *
≥9 h		0.81 (0.67–0.98) *		0.87 (0.71–1.05)
Depressive mood (vs. No)				
Yes		0.93 (0.83–1.05)		0.97 (0.86–1.09)
Cognitive function score (vs. <10)				
≥10		1.55 (1.39–1.74) ***		1.29 (1.14–1.46) ***
**Upstream Factors**				
Place of residence (vs. Urban areas)				
Rural areas			0.59 (0.52–0.66) ***	0.61 (0.54–0.69) ***
Educational level (vs. 0)				
6 years			1.21 (1.05–1.39) **	1.13 (0.98–1.31)
9 years			1.49 (1.24–1.79) ***	1.33 (1.09–1.61) **
≥12 years			2.27 (1.82–2.83) ***	1.98 (1.58–2.49) ***
Personal income (vs. No)				
Yes			1.11 (0.99–1.24)	1.11 (0.99–1.25)
Health education (vs. No)				
Yes			1.30 (1.17–1.45) ***	1.29 (1.16–1.44) ***
Hosmer–Lemeshow test	*χ*^2^ = 12.52 *p* = 0.129	*χ*^2^ = 10.08 *p* = 0.259	*χ*^2^ = 3.41 *p* = 0.906	*χ*^2^ = 6.47 *p* = 0.595

Note: * *p*-value < 0.05, ** *p*-value < 0.01, *** *p*-value < 0.001. *OR*, odds ratio; CI, confidence interval; BMI body mass index; ADL, activities of daily living. Model 1 was the crude model. Model 2 adjusted for the potential predictors (Current drinking, Regular exercise), Model 3 adjusted for the potential predictors (Marital status, Medical insurance), Model 4 adjusted for the potential predictors (Current drinking, Regular exercise, Marital status and Medical insurance).

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
