# Peer review of "Prevalence and Associated Factors of Compliance Behaviors among Middle-Aged and Older Hypertensive Patients in China: Results from the China Health and Retirement Longitudinal Study"

_ijerph, 2020, doi:10.3390/ijerph17197341_

Round 1
Reviewer 1 Report
Hypertension (HTN) is a global concern. This manuscript provided information regarding the trend and influencing factors on HTN management in China. Some similar studies had been published as cited in reference 11 & 34. The author should further describe the importance of this study and the differences to previous studies. Also, please check the terms ‘adherence’ and ‘compliance’ as they were slightly different in definition.
I have several comments in each section of the manuscript.
For the introduction:
- The authors described an increasing incidence of HTN in younger population. Please provide related information such as age group and percentage with reference.
- Sullivan et al study was cited to indicate 20%-40% of re-hospitalization were related to poor compliance behavior. However, the study was published in 1990, 30 years ago. It was unclear if the recent trend was the same as 30 years ago.
- Compliance to lifestyle modifications and medications were highlighted in HTN management guidelines. It would be better to describe why the compliance to lifestyle modifications was not included for analysis.
- What is the common method used to assess medication compliance in the reviews? How sensitive of the method to the medication compliance?
- Blood pressure (BP) monitoring was proposed in HTN education guidelines. What is the effectiveness of regular BP monitoring on BP control?
- As stated, previous studies found poor compliance behaviors in China. The authors only stated 12%-38.5% of participants would monitor the blood pressure regularly. How about the rate of other compliance behaviors in China?
For the methods:
- The inclusion criteria of aged 45 or above was not clear.
- How to categorize the variables into downstream, midstream and upstream?
- Please state clear if HTN was excluded in the number of complications.
- Regarding the term ‘current smoking’, the DA059 might not be well interpreted. Answer of ‘yes’ to ‘have you ever smoked’ might not represent current smoking only. The reply included those smoked before, but quitted now.
- The authors defined regular exercise as 5-7d/week for >=10min/d. The definition is slight from HTN guidelines that the duration should be at least 30min/d. What are the considerations to change the criteria?
- For the educational level, I suggested to present in years, e.g. elementary school = 6 years.
The title of Table 3 needs to be revised in result section.
For the discussion:
- The sentence ‘Most of the participants (9.9%) only measured blood pressure 3 times/year’ was confusing as most of them (40.7%) measured BP >= 12 times/year.
- In addition to strengthen the health education on BP monitoring, how about the accessibility of BP measurement in each household or individual?
- Participants with more than 3 complications had a better compliance. What was the most common reported complication among the participants? A further discussion was suggested.
- There were 2573 participants having cognitive impairment, score <10. How would you validate their responses? Did their responses influence the result of this study?
- The conclusive thought of ‘better cognitive function were more likely to comply with blood pressure monitoring’ was unclear. The dichotomous choice of cognitive function could not reach the conclusive thought.
- The authors suggested the rural population had relatively low levels of income, education etc. Is there any statistical significant at baseline between rural and urban population? If so, how do these factors influence the compliance behaviors?
- On page 9 line 289, please check if it should be 1.97 or 1.98 from Table 2.
- The suggested targeted health education at individual level was too broad. What kind of health education was having among the participants? Does the content of health education influence the compliance behaviors?
- It was unclear about the suggestions of medication burden and accessibility of medication at national policy level as they were not discussed in previous sections.
Author Response
We have carefully revised the whole manuscript point by point according to each comment of the reviewers.Please see the attachment.

Reviewer 2 Report
The article concerns an important scientific problem, and the adopted research procedure is correct. I evaluate the article positively, but I suggest introducing a few changes that will increase its value. Here are the detailed comments:
The introduction does not specify the research gap that the authors are trying to bridge in the article.
The discussion should try to explain the role of individual independent variables in shaping the dependent variable. This is especially true of unusual test results.
In the final part of the work, there is no indication of the contribution of the research to the science system and the directions of further research postulated by the authors.
Author Response

(The authors gave the same response as above.)

Reviewer 3 Report
First of all, I would like to congratulate the authors for this work. The subject seems very interesting to me since the aging of the population is a fact that is increasing throughout the world. By increasing life expectancy, the incidence of chronic diseases such as hypertension also increases. Therefore, to preserve the health of our elderly, it is important to achieve good adherence to treatments.
In general, the work seems quite correct to me, but nevertheless I would like to clarify some aspects with the sole purpose of improving the quality of the manuscript. I detail it below:
- On lines 73-74 there is a sentence that is not referenced. If not referenced, it may seem like speculation.
- At the beginning of the material and method section, I would dedicate a sentence to explain what type of study it is. In this case it would be a descriptive cross-sectional study
The methodology seems adequate to me, the results are clearly expressed and the conclusions are consistent with the results.
Regards
Author Response

(The authors gave the same response as above.)

Reviewer 4 Report
Comments to the authors
The study aimed to explore the proportion of Chinese patients with hypertension reporting compliance with prescribed antihypertensive therapy and BP monitoring. These issues are very important in the management of hypertension. The study included a large sample size and provides exploratory analyses of factors determining compliance. However, there are several issues that limit the strength of this work:
- Participants were classified as hypertensives on the basis of self-reporting – many of those classified as hypertensives may not be truly hypertensives and several others who were excluded from this analysis may be truly hypertensives.
- Compliance was also assessed on the basis of self-reporting; self-reported compliance may differ considerably from true compliance. The study did not include an objective method to assess compliance with treatment.
- The study does not even report the average number of antihypertensives administered to these patients – there in no report of pharmaco-epidemiological data.
- Several factors that determine compliance were not assessed in this analysis. For example, severity of hypertension is an important determinant. The complexity of antihypertensive regimen is another major determinant.
- Compliance to blood pressure monitoring was also not objectively assessed. Blood pressure monitoring at home? Blood pressure monitoring at the doctor’s office?
- Were these patients instructed to measure their BP using a prespecified monitoring schedule and provide BP data to the investigator team? If the reviewer understands correctly, compliance was also based on self-reporting.
Author Response

(The authors gave the same response as above.)

Round 2
Reviewer 1 Report
You worked very hard to address my comments. I really appreciated your effort. However, there is a major issue that have not been addressed.
The author might miss the sentences in previous comments: "Some similar studies had been published as cited in reference 11 & 34. The author should further describe the importance of this study and the differences to previous studies." Wen et al (2018) Medication adherence rate among hypertensive patients in mainland China 2006-2016: a meta-analysis [updated in Ref 13]. The article had reviewed 19 articles with 44322 participants and showed an overall medication compliance among Chinese hypertensive patients was 42.5%. A larger sample size was noted in Wen et al. study. The data of current study was collected in 2015 which might be covered in Wen et al. meta-analysis. Why do we need the current study?
In addition, some other issues required further clarification.
1. What does it mean "cognitive score <10"? Why the cut-off is 10? Do those score <10 influence the reliability of the collected data?
2. What is Family-Doctor System? How does it improve the accessibility of blood pressure monitoring?
Author Response
感谢您的评论,请参阅附件。

Reviewer 4 Report
No comments
Author Response
Thank you for your support in this study.
